# Analyzing E-Bikers’ Risky Riding Behaviors, Safety Attitudes, Risk Perception, and Riding Confidence with the Structural Equation Model

**DOI:** 10.3390/ijerph17134763

**Published:** 2020-07-02

**Authors:** Tao Wang, Sihong Xie, Xiaofei Ye, Xingchen Yan, Jun Chen, Wenyong Li

**Affiliations:** 1School of Architecture and Transportation, Guilin University of Electronic Technology, Jinji Road 1#, Guilin 541004, China; wangtao_seu@163.com (T.W.); sihongxie_guet@163.com (S.X.); traffic@guet.edu.cn (W.L.); 2Faculty of Maritime and Transportation, Ningbo University, Fenghua Road 818#, Ningbo 315211, China; 3College of Automobile and Traffic Engineering, Nanjing Forestry University, Longpan Road 159#, Nanjing 210037, China; xingchenyan.acad@gmail.com; 4School of Transportation, Southeast University, Dongnandaxue Road 2#, Jiangning Development Zone, Nanjing 211189, China; chenjun@seu.edu.cn

**Keywords:** electric bike riders, traffic safety, SEM, risky riding behaviors

## Abstract

To identify and quantify the factors that influence the risky riding behaviors of electric bike riders, we designed an e-bike rider behavior questionnaire (ERBQ) and obtained 573 valid samples through tracking surveys and random surveys. An exploratory factor analysis was then conducted to extract four scales: riding confidence, safety attitude, risk perception, and risky riding behavior. Based on the exploratory factor analysis, a structural equation model (SEM) of electric bike riding behaviors was constructed to explore the intrinsic causal relationships among the variables that affect the risky e-bike riding behavior. The results show that the relationship between riding confidence and risky riding behavior is mediated by risk perception and safety attitudes. Safety attitude was found to be significantly associated with risky riding behaviors. Specifically, herd mentality is most closely related to safety attitudes, which means that those engaged in e-bike traffic management and safety education should pay special attention to riders’ psychological management and education. Risk perception has a direct path to risky riding behaviors. Specifically, stochastic evaluation and concern degree are significantly related to e-bike riders’ risk perception. The findings of this study provide an empirical basis for the creation of safety interventions for e-bike riders in China.

## 1. Introduction

E-bikes are electric-assist bikes that can travel longer distances and provide higher mobility and faster speeds to users compared to conventional bicycles. E-bikes have gradually become the most popular commuting tool for urban citizens in developing countries, especially in China, Vietnam, and Thailand [1]. By the end of 2018, the ownership of e-bikes in China exceeded 250 million, increasing 20% per year [2]. The rapid increase in e-bike ownerships has created serious safety issues. E-bike-related crashes significantly increase every year, especially fatal crashes. According to an annual report on traffic accidents in China, the total number of road traffic accidents in China between 2013 and 2017 was 56,200. In 2008, approximately 5107 e-bike riders died in traffic crashes, accounting for 5.4% of the total fatal crashes. In 2016, approximately 7201 e-bike riders died in road accidents, accounting for 8.9% of the total fatal crashes [3]. Evidently, e-bikes have played a major role in the increase in road traffic injuries and deaths [4]. Therefore, a growing number of studies have focused on the safety issues related to e-bikes [5]. In China, an e-bike is defined as an electric two-wheel vehicle with relatively low speeds and low weight compared to motorcycles. E-bikes can be classified into three types: pedal, light motor, and motor, with maximum designed speeds of 25, 50, and more than 50 km/h, respectively. Due to the lack of current management standards, the dynamic performance of pedal-type and light-motor-type e-bikes is not very different. Due to new regulations, high-speed motor-type e-bikes have become almost nonexistent in China. Therefore, we mainly focused on pedal-type and light-motor-type e-bikes.

Due to the massive increase in injuries and deaths related to e-bike crashes, improving road safety for e-bike riders is important for transportation engineers. Numerous studies have indicated that human risk behavior factors are the core cause of the most traffic crashes. Analogously, the risky riding behavior of e-bike riders is an essential component of traffic safety management research and practice. Essentially, risky riding behavior is often manifested as lacking a safety-cautious attitude, blind riding confidence, a weak awareness of risky riding behavior, and violating traffic rules, all of which can result in high e-bike accident rates and casualties. Taking the violation of traffic rules as an example, riders usually run red lights, and use vehicle lanes and sidewalks. All these illegal behaviors result in the frequent occurrence and higher severity of traffic crashes. Risky riding behaviors are the external manifestations of the subjective psychological characteristics of riders. Most studies have focused on the relationships between factors influencing risky riding behavior including vision, age, maneuvering capability, transcendence, and distraction. The existing studies have not considered risk perception, risk attitude, riding confidence, and their exogenous indicators. Therefore, studying the deep-seated causes of risky e-bike riding behaviors can provide a theoretical basis for the intervention and control of risky behaviors, which has important practical implications.

In terms of the methods used to study these deep-seated causes, traditional statistical analysis methods poorly explain the measurement errors of latent variables; thus, some researchers have used structural equation models (SEMs) to estimate latent variables. Although this method has some limitations, such as its inability to clearly reveal the influence of the omitted variables, its difficulty in dealing with nonlinear problems, and its inability to compensate for limitations in study design and method [6], it performs well for measuring potential influencing factors and evaluating structural relationships [7]. Thus, SEMs have also been widely used in the study of deep-seated causes of driving behavior.

To explore the internal formation mechanism of risky e-bike riding behavior, this paper proposes an SEM using four scales: riding confidence, risk perception, safety attitude, and risky riding behavior. These scales are abstract concepts of social and behavioral sciences. Risk perception, riding confidence, and safety attitude reflect riders’ psychological feelings, which are directly or indirectly related to risky riding behavior. This study involved the collection of data regarding individuals’ riding behaviors on e-bikes and their intentions through questionnaires. Based on the SEM theory, the relationship model between risky riding behavior and the associated intentions was established. This empirical research on the relationships provides theoretical support for risk education and management. We addressed the following research problems:(1)What are the significant relationships among risky riding behavior, riding confidence, safety attitude, and risk perception, and their secondary exogenous variables?(2)How can strategies be implemented for intervening and preventing the risky riding behavior and understand the psychological mechanism of e-bike riders based on the SEM results?

## 2. Literature Review

When studying driving behavior, domestic and foreign scholars have mainly focused on motor vehicles. The existing research on motor vehicle driving behaviors was mostly conducted from the perspective of perception, personality, and attitudes. Ulleberg et al. [8] first studied the relationship between personality characteristics, attitudes, and social perception factors in young drivers in Norway, finding that personality indirectly influences behavior by influencing individuals’ perception of the environment. Jing et al. [9] explored the interaction of driving attitude, personality, and other factors on aberrant driving behaviors in China. Zhao et al. [10] identified the relationships between driver’s demographic characteristics, attitude characteristics, and driving behaviors. The traffic safety climate has gradually been applied to the study of driving behavior [11,12]. Zhang [11] demonstrated that the traffic safety climate can be used as a mediator variable in the influence of personality on driving behavior. A wealth of literature has confirmed the relationship between safety attitude, risk perception, personality, and the driving behavior of motor vehicles. However, driving confidence is rarely mentioned in the study of driving behavior.

Psychological factors, such as safety attitude and risk perception, are also considered to be significantly related to the safety of two-wheeled riders [13]. Chen et al. [14] explored the relationships between personality factors, attitudes toward traffic safety, and risky riding behaviors among young Taiwanese motorcyclists. Sukor et al. [15] found that attitude and perceived behavioral control are significantly related to motorcyclist safety. For e-bikes, the literature has focused on external characteristics related to e-bike riding behaviors, including demographic variables, riding skills, riding distraction, traffic characteristics, and management measures [16,17,18]. In a comparison study of e-bikes and conventional bicycles, e-bike riders were found to commit more violations and experience more hazards than conventional cyclists [19]. Some studies demonstrated that e-bike riders are more likely to crash than conventional cyclists [20]. However, the internal formation mechanism of risky e-bike riding behaviors has rarely been examined. Yao et al. [21] demonstrated the relationship between aberrant riding behavior, traffic safety attitude, and risk perception in Chinese e-bike riders. Then, Zheng et al. [22] found that risky riding behavior is not only related to risk perceptions and safety attitudes, but also to personality traits among Chinese e-bikers. Although previous research on e-bikes has demonstrated the validity of risk perceptions and safety attitudes in predicting unsafe riding, the role of riding confidence in unsafe riding behavior is still unclear. The e-bike riding behavior scale has mainly been based on the research on motor vehicles and motorcycles in the literature, but specific behaviors that are more important for e-bike riders, such as leading behavior, have been less studied. Thus, a more comprehensive model must be built to explore the relationship between psychological factors and the unique riding behavior of e-bike riders.

Early research on the relationship models in the traffic field mainly used traditional statistical analysis methods such as probit, logistic [23], and network modeling methods, including artificial neural networks (ANNs) and support vector machines (SVMs) [24]. As the number of variables involved in the studies increased and the complex relationship between multiple variables needed to be described, researchers introduced the SEM into the field of traffic behavior research. SEMs can simultaneously handle a large number of endogenous and exogenous observed variables, which is especially suitable for the processing of multiple variable relations that cannot be accurately and directly measured in sociological and psychological research [25]. SEM is a comprehensive analysis method in statistical analysis. Its idea originated from the concept of path analysis proposed by geneticist Wright in the 1920s and introduced into the study of latent variables, which was combined with the factor analysis method to form the SEM method. With the development and improvement of SEM methods, some derivatives have been developed. To overcome some limitations of the traditional SEM, the constrained autoregression–structural equation model (ASEM) was developed to handle omitted variables and measurement errors [26]; the ANN was introduced to solve interaction and nonlinear problems [25]; the Bayesian method and the SEM were combined to solve the problem of sample size [27]; a new descriptive fit measure, the homoscedastic fit index (HFI), was proposed to detect omitted nonlinear terms in the SEM [28]; and a hybrid three-stage SEM-ANN-interpretive structural modeling (ISM) approach was introduced to further strengthen the study of the relationship between the factors [29]. However, the existing research on driving behavior has mainly used the traditional SEM, and its derived model has seldom been studied.

## 3. Materials and Methods

### 3.1. Respondents and Procedure

To ensure that the sampling error meets a certain confidence level, the sample size in the random sampling should be determined. Considering the huge number of e-bikes, it can be regarded as infinite without violating the reasonable conditions. Therefore, the simple random sample size of statistics was calculated using the following equation:(1)$$n=\frac{z^{2}p(1-p)}{e^{2}}$$
where n is the required sample size, z is the standard score, e is the allowed sampling error range, and p is the estimated probability of target data. With a 95% confidence interval, the z-value was set to 1.96, and the e was a 5% error range. The p-value was 0.05 according to the maximum absolute error estimate.

We calculated that at least 385 questionnaires were needed. The preliminary questionnaire survey was administered to 60 individuals with e-bike violations in Guilin, a city in Southern China, as well as some preparatory drivers studying at a driving school. The preliminary questionnaire survey was designed to test whether the meaning and description of the questionnaire could be correctly interpreted by the subjects to avoid bias due to misunderstandings. The reliability of the questionnaire survey was assessed with SPSS (IBM company, Chicago, Illinois, USA), and the items with an acceptable level of reliability (Cronbach’s α > 0.7) were retained [30]. Some of the retained questions were restructured and revised to design the final questionnaire.

The random and tracking surveys were applied in the formal questionnaires. For the random surveys, e-bike riders were selected anonymously in Nanning and Guilin, and a total of 434 questionnaires were collected. After excluding those with incomplete and missing answers, a total of 381 valid questionnaires remained. There were 236 valid questionnaires in Nanning city and 145 valid questionnaires in Guilin city. For the tracking surveys, with the assistance of the traffic police, 192 valid questionnaires were collected among the e-bike riders who had experienced traffic crashes. There were 46 crashes without injuries, 134 with minor injuries, and 12 with serious injuries. The effectiveness rate of the questionnaire was 87.7%.

A total of 573 valid questionnaires were obtained from the random surveys and tracking surveys, and finally, the questionnaires were recorded uniformly and converted into analyzable data formats for further statistical analysis.

### 3.2. Measurements

#### 3.2.1. Riding Confidence

The riding confidence scale, which consisted of four questions in total, was based on the research of Wong et al. [31]. The riding confidence scale measured the riders’ judgment of their riding skills (e.g., “In an unfamiliar road environment, I can handle any emergency”). However, according to the definition of confidence in social psychology, the source of confidence is not only the affirmation of technical capacity but also the affirmation of individual judgment, and especially the judgment of the behavior and surrounding environment based on the individual’s riding experience (e.g., “I can accurately determine the vehicle’s movements”).

Therefore, we added four questions based on Wong’s riding confidence scale. The riding confidence scale was divided into two dimensions to evaluate technical capacity confidence and environmental judgment ability confidence. Each question was measured by a five-point Likert scale. On this scale, 1 through 5 represented totally disagree, disagree, partially agree, agree, and totally agree, respectively. The higher the score, the higher the confidence of the respondents.

#### 3.2.2. Risk Perception

The risk perception scale was adapted from the study of Rundmo et al. [32]. Risk perception is defined as an individual’s subjective perception of potential danger. Thus, individual risk perception precedes risky riding behavior. Existing research has shown that risk perception can be measured through emotional concerns and cognitive-based evaluation [33].

Rundmo reported the worry and concern scale based on emotion, which contained six traffic crashes, traffic injury worries, and multiple anxiety-inducing items (e.g., “I often worry I will get hurt in a traffic crash”) [32]. Machin’s danger perception scale is based on a perception evaluation with six questions, including the subjective perception of the possibility of a crash and subjective perception of the degree of a crash consequence [33]. The subjective perception of a crash possibility was ascertained by asking respondents to answer the question, “How likely do you think you are to be in a traffic crash because of a vehicle?” The subjective perception of the degree of a crash consequence was measured by asking respondents to evaluate the risk of running a red light, drinking and riding, and other risky behaviors.

Therefore, we designed the risk perception scale, which was divided into two dimensions to evaluate the degree of concern and the danger perception of crashes. Each question was measured by a five-point Likert scale. The higher the score, the more worried and anxious the rider is about the risk, the higher the degree of danger of the subjectively perceived risk behaviors, and the more serious consequences of the perceived accident.

#### 3.2.3. Safety Attitude

The safety attitude scale was adapted from Rundmo [32]. This scale consists of 10 questions and is divided into two dimensions: the attitude of personal responsibility in traffic safety and the attitude toward traffic regulations. According to the characteristics of the crowd line of the e-bike riders, we constructed four additional questions to evaluate the degree of individual dependence on reference groups, named “herd mentality” (e.g., “I will not be blamed for breaking the rules by the people next to me”).

Therefore, we designed the safety attitude scale, which was divided into three dimensions to evaluate e-bike riders’ attitudes regarding personal responsibility, traffic regulations, and herd mentality. Each question was measured by a five-point Likert scale. The higher the score, the more positive the attitude toward safety, the stronger the sense of obligation to abide by the traffic rules, and the lower the preference of herd mentality.

#### 3.2.4. Risky Riding Behavior

Yao compiled an e-bike riding behavior questionnaire (ERBQ) based on the motorcycling behavior questionnaire and cycling behavior questionnaire, combined with China’s national conditions and e-bike characteristics [21]. The ERBQ compiled by Yao mainly examines e-bike riders’ aggressive behavior, error, irregularity, general behavior, impulsive behavior, and speeding behavior. However, negligence behavior questions were not included in the ERBQ [8], which may be an e-bike rider’s most important risky behavior, since it hinders them from receiving adequate riding training.

Therefore, referring to previous questionnaires on motor vehicles, motorcycles, and bicycles, and combining them with observations on e-bike riding behaviors, we designed an e-bike riding behavior questionnaire (ERBQ). The ERBQ was used to measure violation, negligence and error, aggressive behavior, and leading behavior. Each question was measured by a five-point Likert scale. The higher the score, the higher the frequency of the risky riding behavior.

#### 3.2.5. Crash Involvement and Demographics

Crash experience includes the frequency of crashes over nearly three years, including crash severity (which, from high to low, was scored as: 3, needing personnel for medical treatment; 2, causing minor injuries; and 1, causing property damage) and other multi-item indicators.

Other information included demographic information (sex, marital status, age, domicile of origin, etc.), social and economic characteristics, level of education, the type of e-bikes, the main purpose of travel, and riding background (motor vehicle driving experience, e-bike riding time, frequency, and distance).

## 4. Results

### 4.1. Respondent Characteristics

The survey respondents all had e-bike experience. Of the 573 respondents, 328 men accounted for 57.2% of the test population, while 245 women accounted for the remaining 42.8%. The respondents ranged in age from 14 to 63 years, with an average age of 29.7 years, and the youth (age less than 35 years old) accounted for 67.5% of the respondents. Notably, there are no mandatory restrictions on the age of e-bike riders in China, so riders under the age of 16 years were found. In terms of marital status, 253 married people accounted for 44.2% of the total, whereas 320 unmarried people accounted for the remaining 55.8%. In terms of education, 244 respondents had earned a college degree or above, accounting for 42.6% of the test population, and 183 people had a high school or secondary school degree, accounting for 31.9% of the test population; the remainder had education at or below the junior high school level.

Most of the e-bike riders were not experienced with motor vehicle driving: 196 e-bike riders had some experience with motor vehicle driving, accounting for 34.2% of the test population, and 383 e-bike riders had no motor vehicle driving experience, accounting for 66.8%. Among the respondents, 431 rode light-motor-type e-bikes, a far higher number than those who rode a pedal-type ordinary e-bike, the proportion of which was 24.8%.

In terms of riding experience, the average riding experience was 3.8 years, with riding experience ranging from at least 0.5 years and up to 17 years. After a cross-comparison study, some respondent stated that they started riding their e-bikes before reaching 12 years of age. The respondents covered an average distance of 12.2 km per day, and the standard deviation was 10.7 km; the results showed that there was significant variability in the distance measurement, which might be caused by the differences in the main purpose of e-bike usage. The average weekly use of e-bikes was 4.78 days, and the standard deviation was 2.0 days, indicating a high frequency of use. When evaluating the purpose of travel, 69.1% of respondents used them for commuter activities, such as riding to work or school; 23.9% used them for daily routines; and the remaining 6.8% used them for freight and other business activities.

Over the past three years, 16.6% of respondents had been punished or warned for violating the rules and regulations. Of the 264 respondents (46.0%) who had experienced traffic crashes, 81 (14.1%) involved property damage, 164 (28.6%) resulted in minor injuries, 19 (3.3%) resulted in severe injuries necessitating medical intervention, and 108 respondents (18.8%) claimed primary responsibility for the crash. Table 1 shows the basic demographic information of the respondents.

### 4.2. Exploratory Factor Analysis

A total of 300 questionnaires were randomly selected for exploratory factor analysis (EFA) to explore the underlying structure for variables measured by multiple items. For the EFA process, we adopted the principal axis and varimax rotation approaches. Based on the EFA results, the main factors of riding confidence, risk perception, safety attitude, and risky riding behavior were extracted. Some items were moved or removed for low loading value or cross-loading. For riding confidence, two dimensions were determined, technical capacity and judgment ability, explaining a total of 76% of the total variance, consistent with the preliminary scale design. For risk perception, the “worry and concern” dimension based on emotion in Rundmo’s study was not complete; the factor analysis results showed that it should be divided into two dimensions. Therefore, three dimensions were determined, danger level, concern degree, and stochastic evaluation, explaining a total of 64% of the total variance. For safety attitudes, three dimensions were determined, safety responsibility, traffic regulation, and herd mentality, explaining a total of 71% of the total variance, consistent with the preliminary scale design. For risky riding behavior, four dimensions were determined, negligence and error, rule violation, aggressive behavior, and leading behavior, explaining a total of 68% of the total variance, consistent with the preliminary scale design. All potential variables showed an acceptable level of reliability (Cronbach’s α > 0.7). The factor loading and Cronbach’s α of the final items are shown in Appendix A.

### 4.3. Structural Equation Model Testing

#### 4.3.1. Theoretical Model Hypothesis

The SEM method is composed of a measurement model and a structural model. SEM is a type of verification model that is mainly used to verify the reliability of scientific assumptions regarding the basic structure of a theoretical model. It can simultaneously handle many endogenous and exogenous variables by establishing the relationships between variables based on linear combinations [34]. The measurement model mainly represents the relationship between measurement variables and latent variables. The formulation of the model is as follows:(2)$$X_{m}=A_{x}\xi +\delta$$
(3)$$Y_{n}=A_{y}\eta +\varepsilon$$
where $X={\left(x_{1},x_{1},x_{3},\ldots ,x_{m}\right)}^{T}$ is a column vector composed of m exogenous indices, $\xi ={\left(\xi_{1},\xi_{2},\xi_{3},\ldots ,\xi_{u}\right)}^{T}$ is a column vector composed of u exogenous latent variables, $A_{x}$(m×u) is the factor loading matrix of X on ξ, and δ is the error of the exogenous variables. Similarly, Y is the vector composed of endogenous indices, η is the vector composed of endogenous latent variables, $A_{y}$ is the factor loading matrix of Y on η, and ε is the error of the endogenous variables.

The structural model represents the relationship between exogenous latent variables and endogenous latent variables. The formulation of the model is as follows:(4)η=Bη+Γξ+ς
where B=(m×m) is the coefficient parameter matrix of the endogenous latent variables’ vector η, Γ=(m×n) is the coefficient parameter matrix of the exogenous latent variables’ vector ξ, and ς is the residual vector and represents the unexplainable part of the model.

However, the initial path relationships of the theoretical model are unclear. Accordingly, we first employed correlation analysis to preliminarily judge and assess the relationships between the factors, which provided the basis for the hypothetical structural equation model paths. Assuming full path relationships between the four scales, and according to the parameter estimation results of the model to adjust the paths of the variable intrinsic relationships, the full sample (*N* = 573) data were used for the correlation analysis and validation analysis. Based on the results of correlation analysis and factor analysis, the SEM with riding confidence, safety attitude, risk perception, and risky riding behavior was constructed. The correlation analysis demonstrated that the riding confidence was significantly correlated with risk perception, safety attitude, and risky riding behavior (*p* < 0.05); riding confidence was identified as a potential exogenous variable. Finally, according to the relevant analysis results, six hypothetical analysis paths were determined: riding confidence → risky riding behavior, riding confidence → risk perception, riding confidence → safety attitude, risk perception → safety attitude, risk perception → risky riding behavior, and safety attitude → risky riding behavior.

The SEM constructed in this study included two-layer factor structures: the 12 dimensions, such as danger level, were the initial factors, and the four scales, such as risk perception, were the higher-order factors. We took the 12 initial factors as variables for factor analysis in the preliminary stage of evaluating the higher-order potential variable, and the multi-items of each initial factors as observable variables. According to the hypothesis of the six analysis paths, the risky e-bike riding behavior theoretical model was developed, as shown in Figure 1. In the theoretical model, the observable variables were not listed in order to simplify the model expression. For higher-order potential variables to be estimated, a factor path of each higher-order potential variable was set to one as the reference index [35].

#### 4.3.2. Goodness of Fit and Estimated Results

An adaptive analysis of the proposed theoretical model was conducted to verify whether the established path relationships were valid. Table 2 shows the fit index results of the model. The results showed that the chi-squared value (CMIN) of the preliminary theoretical model and data fitting was 257.266 (*p* < 0.001), the degree of freedom (DF) was 68, and CMIN/DF was 3.378, which met the standard (between 1 and 5). RMSEA was 0.107, higher than the highest standard (0.05). This result showed that the gap between the theoretical model and the saturated model did not meet the standard. The normed fit index (NFI), comparative fit index (CFI), and increasing fit index (IFI) were 0.846, 0.879, and 0.881, respectively, which did not meet the evaluation standard of 0.90. The goodness-of-fit index (GFI) was 0.873, which did not meet the recommended standard of 0.90 [36]. The theoretical model required further revision because the overall fit of the model was not within an acceptable range.

The results of the path analysis showed that some paths had no significant relationship in the overall structural equation model. First, the standardization factor load between the initial potential variable “danger level” and the higher-order potential variable “risk perception” was only 0.314 (preferably between 0.4 and 0.96), so the variable “danger level” could be deleted. Second, the test results of the regression coefficients showed that the direct path between “riding confidence” and “risky riding behavior” was nonsignificant in the tested model (CR (Constituent reliability) = 1.301, *p* = 0.193), so this path was also deleted. The SEM with path coefficients and load coefficients is shown in Figure 2.

The relationship between the error items was established, the original independent restriction was released, and one pair of parameters was modified each time the parameters were released. After revising the model, the maximum likelihood estimation method was used to test the overall suitability of the model. Table 3 shows the fit index results of the revised model. The test results showed that the CMIN was 155.295 (*p* = 0.056), DF was 69, and CMIN/DF was 2.250, conforming to the standard. RMSEA was 0.028, which met the standard of 0.05, indicating that the gap between the theoretical model and saturated model met the standard. The coefficient of determination (*R*^2^) was used to measure the variation ratio of the dependent variable that can be explained by independent variables to judge the explanatory power of the model. The values of *R*^2^ were 0.41, 0.67, and 0.74, respectively, indicating that the explainable levels of independent variables reached a medium or high level. The variables $e_{1},e_{2},\ldots ,e_{12}$ and ${er}_{1},{er}_{2},{er}_{3}$ represent the structure residuals. Each structure residual corresponds to a potential variable, which reflects the unexplainable variances of the structure equation. The standardized value of residuals was below 2.58, indicating that the internal structure of the model had an adequate fit degree.

The NFI, CFI, IFI, and GFI were 0.913, 0.922, 0.925, and 0.948, respectively, all of which met the recommended standard of 0.90. The index results indicated that the revised structural equation model in this paper had an adequate fit for the observable data. The revised structural equation model with path coefficients and load coefficients is shown in Figure 3.

## 5. Discussions

### 5.1. Analysis of Direct, Indirect, and Total Effects

After the verification and correction of the model, the paths were tested. The test results are shown in Table 4. The results showed that the six direct paths of the model hypothesis, except for the path between riding confidence and risky riding behavior, reached the 95% significance level. According to the normalized path coefficients and load coefficients of each component, the results of the model analysis were discussed.

Safety attitude is directly related to risky riding behavior. As seen from the SEM, the overall utility of the variables on the safety attitude scale was, from large to small: herd mentality (0.83), traffic regulations (0.61), and safety responsibility (0.75). The consequence of the factor load coefficient indicated that e-bikers’ herd mentality has the most significant influence on their safety attitudes. This result is different from the study of driving behaviors of motor vehicle drivers, which showed that the association between traffic regulations and safety attitudes is the most significant. Therefore, more attention should be paid to riders’ psychological management and safety education. The results showed that e-bike riders that have a greater sense of responsibility for safety, a greater sense of obligation to obey traffic rules, and a lower preference to herd others are less likely to engage in risky riding behaviors.

Risk perception and risky riding behavior have significant direct and indirect path relationships. The questionnaire defined “danger level” based on the respondents’ understanding of the assessment. According to feedback from the respondents, the mean value of the danger level was 4.37 (SD = 1.09), showing a higher level of danger perception of the risk behavior of respondents. However, as the danger level increased, the path relationship between risk perception and risky riding behavior in the whole structural equation model did not significantly change, indicating that danger level is not a good explanation for the risk perception scale. The overall utility of the variables “concern degree” and “stochastic evaluation” were 0.85 and 0.78, respectively, on the risk perception scale, showing higher load values. Accordingly, we concluded that the problem is not that e-bike riders think their riding behavior is dangerous, but that they believe there is a lower probability of being in danger than others in a general sense. This result showed that e-bike riders who worry more about riding and think there is a higher possibility of being in danger are less likely to display risky riding behaviors. E-bike riders who scored high on risk perception tended to have positive attitudes toward traffic safety, and thus tend to have safe riding behaviors.

Riding confidence is not significantly associated with risky riding behavior, but it could be indirectly related to riding behavior by using risk perception and safety attitude as intermediaries. From the perspective of the variables of the riding confidence scale, confidence involved two dimensions: judgment ability (0.90) and technical capacity (0.68). The rider’s judgment ability had more influence on riding confidence than technical capacity. According to the questionnaire, “judgment ability” mainly refers to the riders’ confidence in the judgments of their behavior and the environment, which is accumulated through individual riding experience. For e-bike riders, because traffic management restrictions on riding behavior are not mandatory, the ability to judge one’s accumulated behaviors and the environment with self-affirmation can better explain risky riding behavior. This result showed that e-bike riders who score high on risk confidence tend to have anegative attitude toward traffic safety and risk perception, which in turn causes them to be more likely to display risky riding behaviors.

### 5.2. Prevention and Intervention of Risky Riding Behavior

The SEM analysis results provide some useful information for improving the strategies to ensure safety. The best way to intervene and improve the riders’ intentions is to avoid risky behaviors.

The observed indicator variables of herd mentality, traffic regulations, and safety responsibility in safety attitude had a negative relationship with risky riding behavior, which means that relevant riding safety regulations and education are necessary to prevent risky behavior. Therefore, we recommend a strict enforcement of regulations for e-bikes to decrease traffic accidents. The strong correlation between herd mentality and safety attitude also suggests that herd mentality education and training should be strengthened to improve the awareness of risky behaviors. A rider license penalty point system should be implemented to regulate risky behaviors; for violators, safety knowledge training and examinations should be strictly implemented. By formulating these policies, the public can be shown the importance attached by the government to the safety of e-bikes, and then constantly correct the users’ attitudes toward safe riding.

We also found a significant negative relationship between risk perception and risky riding behavior, and the stochastic evaluation was most closely related to risk perception, showing that the riders have the faulty mindset that traffic crashes will not happen to them. However, traffic accidents are highly unpredictable and no chances should be taken. To increase the awareness of risky behavior and the severity of traffic accidents, education and regulations are key, and must be implemented though a step-by-step process. Gradual and incremental approaches should be implemented to avoid risky riding behavior. For example, uniform riding licenses should be initiated at the first stage, strengthened safety awareness and education should be implemented every year, a mandatory helmet policy should then be regulated to reduce head injuries and fatalities resulting from e-bike and bicycle crashes, and e-bikes with speeds higher than 15 km/h should be forbidden in long-term plans. All of these countermeasures will gradually increase public awareness of risky behaviors. In addition, the media should continue to improve the coverage of traffic accidents, including an in-depth analysis of the causes of traffic accidents to improve public awareness of traffic accidents.

The path relationship between riding confidence and risky riding behavior showed that overconfidence or even blind confidence in one’s riding ability indirectly leads to risky riding behavior. Because e-bikes riding behavior is not constrained compulsively in traffic management, the tolerance of risky behavior increases the confidence of riders and encourages undisciplined riding habits, which increase risky behavior and violations. In addition to laws and regulations, transportation infrastructure such as speed humps, separation between motor lanes and non-motorized lanes, and a proprietary signal phase at intersections with a large volume of e-bikes should be ensured to improve the riding environment and increase benign riding confidence, so that riders with poor riding skills or timid riders can feel that they are in a safe traffic atmosphere, which will reduce the occurrence of traffic accidents.

## 6. Conclusions

In this study, a risky riding behavior scale was designed under China’s national conditions. We used an SEM to evaluate the internal potential relationships between influencing factors and risky riding behaviors. Compared with the traditional statistical methods, the analysis result of the SEM was more accurate; the theoretical model constructed in this study had a better fit as indicated by the various fit indices. The result showed that safety attitude, risk perception, and riding confidence are significantly directly or indirectly related to risky riding behavior. The reported findings provide some useful information for e-bike safety and a theoretical basis for the follow-up research on driving behavior. The research results of the inherent formation mechanism and the relationship among risk riding behaviors provide a theoretical basis for subsequent research.

However, several limitations should be noted. Self-reported data may have some bias with expectations; however, other studies confirmed that the social expectation bias has little impact on participants’ responses. Nevertheless, further studies could be conducted to improve the current questionnaire design to better capture these potential psychological factors. The survey samples were only obtained from two cities, Guilin and Nanning, which cannot accurately represent the whole of China. However, currently, small- and medium-sized cities, especially in southern cities like Guilin, have a higher proportion of e-bike use. According to the comparison of the survey samples between Nanning and Guilin, the influence of geographical difference was not significant. Given the limitations of the SEM, the method should be improved in further research to obtain more accurate results. Since we only focused on e-bikes, it might make more sense to compare different types of two-wheeled vehicles in terms of psychological factors and risky riding behaviors. The risky driving behavior model may be further optimized, and additional considerations will be included, such as driving forgiveness, ecological driving, safety climate, and other aspects, into the model research.

In the future, studies of these relationships need to consider the autonomous vehicles environment [37,38,39]. Cellular automaton models could be used to further refine the interactions of e-bikes and bicycles [40]. These theories can be used to deepen the awareness of e-bike safety and help with the planning and design of non-motorized facilities.

## Figures and Tables

**Figure 1 ijerph-17-04763-f001:**
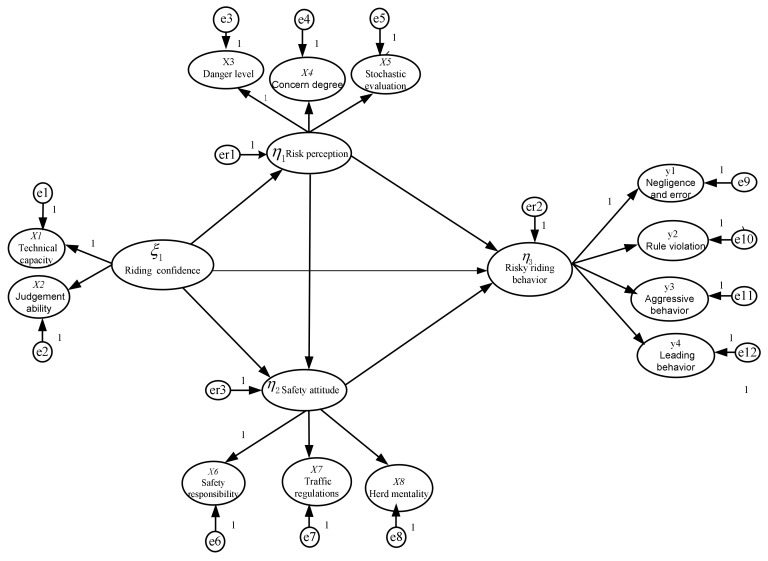
Theoretical structure model of risky riding behavior.

**Figure 2 ijerph-17-04763-f002:**
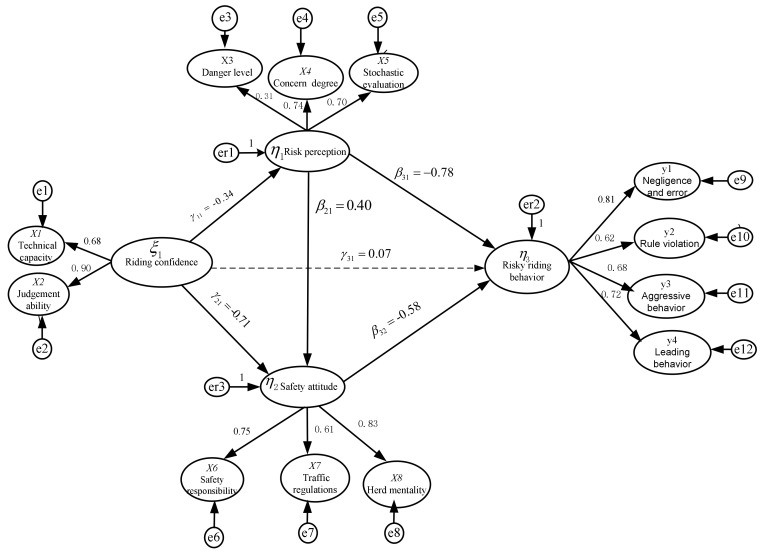
Preliminary fit results of the theoretical model.

**Figure 3 ijerph-17-04763-f003:**
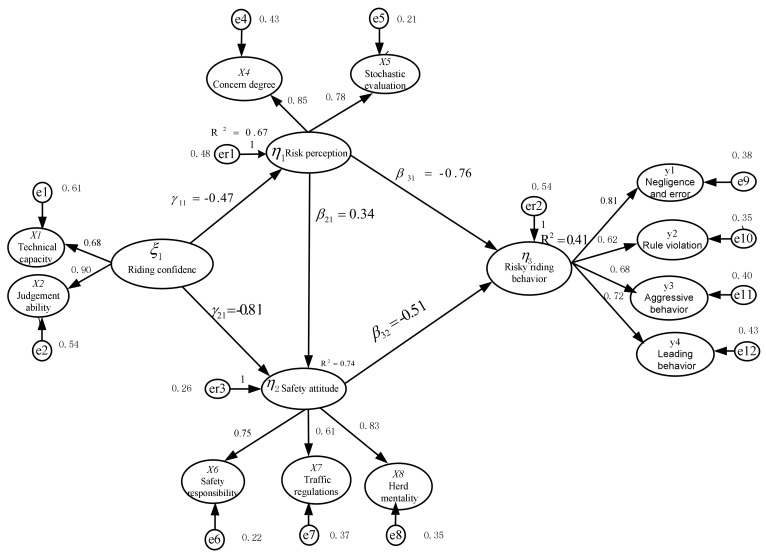
Fit results of the revised model.

**Table 1 ijerph-17-04763-t001:** Summary of respondents’ demographic information (*N* = 573).

Variable	Category	Frequency	Percentage
Sex	Male	328	57.2%
	Female	245	42.8%
Age sections	--	Mean = 29.7, SD = 10.1, Range = 14–63	
	Youth (<35 years)	387	67.5%
	Middle aged group (35–55 years)	161	28.1%
	Seniors group (>55 years)	25	4.4%
Marital status	Unmarried	320	55.8%
	Married	253	44.2%
Educational level	Junior high or below	146	25.5%
	Secondary school	183	31.9%
	College degree or above	244	42.6%
Motor vehicle driving experience	Yes	196	34.2%
	No	383	66.8%
E-bike type	Pedal-type	142	24.8%
	Light motor-type	431	75.2%
Experience in e-bike riding	Riding time (year)	Mean = 3.8, SD = 2.9, Range = 0.5–17	
	Riding frequency (sub-/weekly)	Mean = 4.78, SD = 2.0, Range = 1–7	
	1–2	78	13.6%
	3–5	254	44.3%
	6–7	241	42.1%
	Average riding distance per day (km)	Mean = 12.2, SD = 10.7, Range = 1–42	
	<5	237	41.3%
	5–10	123	21.5%
	>10	213	37.2%
	Main uses		
	Work	303	52.9%
	Go to school	93	16.2%
	Daily routines	137	23.9%
	Business/freight	39	6.8%
Been punished or warned for violations in the past three years	Yes	95	16.6%
	No	478	83.4%
Had a traffic crash in the past three years	Uninjured crash	81	14.1%
	Minor injury crash	164	28.6%
	Severe crash	19	3.3%
	Total	264	46.0%
Had primary responsibility crash in the past three years	Yes	108	18.8%
	No	465	81.2%

**Table 2 ijerph-17-04763-t002:** Comparison of the theoretical model fit index results with the evaluation standard.

Model	Definition	Index Results	Evaluation Standard
DF	Degrees of freedom	68	-
CMIN	Chi-squared	257.266	-
CMIN/DF	-	3.378	1–5
RMSEA	Root Mean Square Error Approximation	0.107	<0.050
CFI	Comparative Fit Index	0.879	>0.900
GFI	Goodness of Fit Index	0.873	>0.900
NFI	Normed Fit Index	0.846	>0.900
IFI	Incremental Fit Index	0.881	>0.900

**Table 3 ijerph-17-04763-t003:** Comparison of the revised model fit index results with the evaluation standard.

Model	Index Results	Evaluation Standard
CMIN	155.295	-
CMIN/DF	2.250	1–5
RMSEA	0.028	<0.050
CFI	0.922	>0.900
GFI	0.948	>0.900
NFI	0.913	>0.900
IFI	0.925	>0.900

**Table 4 ijerph-17-04763-t004:** Direct and indirect effects of riding confidence, risk perception, safety attitude, and risky riding behavior.

Dimension	$\mathrm{Riding}\;\mathrm{Confidence}\;(\xi_{1})$	$\mathrm{Risk}\;\mathrm{Perception}\;(\eta_{1})$	$\mathrm{Safety}\;\mathrm{Attitude}\;(\eta_{2})$
Direct effect
Risk perception ($\eta_{1}$)	$\gamma_{11}$(−0.47)		
Safety attitude ($\eta_{2}$)	$\gamma_{21}$(−0.81)	$\beta_{21}$(0.34)	
Risky riding behavior ($\eta_{3}$)		$\beta_{31}$(−0.76)	$\beta_{32}$(−0.51)
Indirect effect 1
Safety attitude (η2)	γ11×β21(−0.47 × 0.34)		
Risky riding behavior (η3)	γ11×β31(−0.47 × −0.76)	β21×β32(0.34 × −0.51)	
γ21×β32(−0.81 × −0.51)	
Indirect effect 2
Risky riding behavior (η3)	γ11×β21×β32(−0.47 × 0.34 × −0.51)

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
