# Peer review of "Analyzing E-Bikers’ Risky Riding Behaviors, Safety Attitudes, Risk Perception, and Riding Confidence with the Structural Equation Model"

_ijerph, 2020, doi:10.3390/ijerph17134763_

Round 1
Reviewer 1 Report
The authors have done a interesting research to identify and quantify the factors that influence the risky riding behaviors of electric bike riders. There are some comments before this manuscript deemed to be published in IJERPH.
- Conclusions should be condensed;
- Please highlight deficiencies along with the selected type of research;
- Presentation of the figures should be improved;
- Please reduce number of old references
Reviewer 2 Report
This paper evaluates relationship among riding confidence, safety attitude, risk perception, and risky riding behavior of electric bike riders in China, using the structural equation model (SEM). The research topic is worth of investigation and the paper is well written and organized.
I have some minor comments for further improvement:
The strength of the proposed SEM model should be illustrated in the Introduction section. While the SEM may have good performance to measure the potential influencing factors and evaluate the structural relationships, it cannot clearly reveal the influence of the omitted variables and compensate for limitations in study design and method, which are quite common in observational or questionnaire based study. For further reference, authors can see the following paper:
- Tomarken A. J. And Waller N. G. (2005). Structural Equation Modelling: Strengths, Limitations, and Anu. Rev. Clin. Psychol. 2005. 1:31-65
Authors mention that “Considering traditional statistical analysis methods are not good at explaining the measurement errors of latent variables, the structural equation model (SEM) provides the possibility to analyze the relationships between latent and observed variables.” I think author is very harsh in this aspect. Many researchers used latent variable and/or latent class discrete outcome models to address the influence of unobserved variables and that is well established in the literature.
Literature review is well organized. However, the review is only focused on the subject area. Methodological aspect is completely overlooked. Better to incorporate some literatures/ discussions on methodological approaches/ different derivatives used in different relevant studies.
In the Methodology section, the formulation of the SEM model should be introduced in further details, as some interested readers may not be familiar with this method.
Research contribution is pointed out at the end of LR. However, it is not clear enough to follow. Better to elaborate more.
Inclusion of some policy analysis and discussion such as Elasticity of impact; specific safety improvement strategy etc. will give added weightage of the paper.
Reviewer 3 Report
In general: The authors present an interesting study, using sound methodology (Structural equation model (SEM) complemented with exploratory factor analysis (EFA)), to deal with e-bike driver safety.
The new mobility in urban places, will need great research to deal with the involved issues: safety, driver behaviour, compatibility between vehicles, separated infrastructure, surveillance and controls.
Driver behavior comprehension is the core and works like this are very important.
The paper is well structured and the authors deal with the aim of the study and analyse the achievements in the conclusion.
I am grateful to them for this way to write.
However, the paper needs minor revision. Here are my concerns and recommendations:
- Some erratum in text: L109, L120, L351, L413
- Revise text remarked in red : lines 89-91, L95-105, L107-114
- I have some concerns about the representation of the data, knowing the size population of Chinese cities. Which is the proportion of the sample?, which is the sample procedure?, which could be the error?. The authors have to describe the sample selection and the statistical criteria used.
Author Response
Response to Reviewer 3 Comments
First of all, I would like to thank the editor for arranging the review and the reviewer for your valuable suggestions. The authors have carefully answered the questions point-by-point in accordance with the requirements of yours, and made careful modifications to the article, and all of revisions have been clearly highlighted. Because of your suggestions, the revised articles become better and readers can get more valuable information. Thanks again for your help.
Point 1: Some erratum in text: L109, L120, L351, L413
Response 1: The original version of L109, L120, L351, L413 have been corrected in line 114-115,123-125,394-395,469 in the revision version:
Modification:
(1)e-bike riders are found to have higher violation and hazards than cyclists in line 114-115
(2)In addition, the riding behavior scale of e-bike was mainly based on the research of motor vehicle and motorcycle in pervious literature, while the specific behaviors that are more significant among e-bike riders in line 123-125
(3)Direct and indirect effects of riding confidence, risk perception, safety attitude, and risky riding behavior in line 394-395
(4)the separation between motorized and non-motorized lanes in line 469
Point 2: Revise text remarked in red : lines 89-91, L95-105, L107-114
Response 2: The text remarked in red part has been revised to black in the revision version.
Point 3: I have some concerns about the representation of the data, knowing the size population of Chinese cities. Which is the proportion of the sample?, which is the sample procedure?, which could be the error?. The authors have to describe the sample selection and the statistical criteria used.
Response 3: The procedure of sample selection and the statistical criteria have been supplemented in Section 3.1 of the revision version. As shown in line 152-160. And the effective rate of the questionnaire has been supplemented. As shown in line 173-174.
Like other cities in China, the centripetal nature of Guilin and Nanjing’s urban construction and development has led to the formation of high-density land use in the urban center, which has resulted in high-intensity use of e-bike transportation. E-bikes have gradually become the most popular commuting tool for urban citizens and also cause serious safety issue in Guilin and Nanjing like other cities in China. Therefore, it is necessary to study riders’ risky behavior. Through the field survey, data collection of risky behavior was carried out. Finally, 434 questionnaires were sent out. After excluding those with incomplete and missing answers, a total of 381 valid questionnaires remained. There were 236 valid questionnaires in Nanning city and 145 valid questionnaires in Guilin city. Through the assistance of the traffic polices, for the tracking surveys, 192 valid questionnaires were collected among the e-bike riders who had experienced traffic crashes. To maintain the accuracy of the estimations and proper solutions, ensure representativeness, and use multiple observed indicator variables to define latent variables, a much larger and sufficient sample size, from 100 to 200, is recommended when maximum likelihood estimation is used. According to the study [34], a sample size of 400 is adequate for SEM.
This manuscript is a resubmission of an earlier submission. The following is a list of the peer review reports and author responses from that submission.
Round 1
Reviewer 1 Report
lines 27-29 do not make sense. stochastic evaluation may have the greatest impact on risk perception but that is not what is done reading the document.starting close to the end of first paragraph, the sentences are too long and and not easily readable in the introduction. 2nd paragraph is OK but suddenly jumps to SEM at the next paragraph then jump back to data to explain driving behaviors. section 2.1 is OK. Lines 125-126 are confusing and does not add a value to paragraph. Lines 146-148 are confusingly written. Is this based on the Rundmo's paper? when there are mutiple predictors, it is incorrect to use Rsquare (adjusted is more appropriate) to measure the explanatory power. Diagrams are good but you started with somebody else's model or inputs.
Author Response
Dear Reviewer:
Thank you for your comments concerning our manuscript. Those comments are all valuable and very helpful for revising and improving our paper, as well as the important guiding significance to our researches. We have studied comments carefully and have made correction which we hope meet with approval. Revised portion are marked in red in the paper. The main corrections in the paper and the responds to the reviewer’s comments are as flowing:
Point 1: lines 27-29 do not make sense. stochastic evaluation may have the greatest impact on risk perception but that is not what is done reading the document.
Responses1: It really doesn't make sense to just show that the stochastic evaluation has the greatest impact on risk perception, therefore, lines 27-29 were modified to supplement the significance of risk perception influence factors. As shown in the revisions line 29-32.
Modify1: The risk perception had a direct effect on the risky riding behavior; stochastic evaluation and concern degree both had great influences on the risk perception, which meant more efforts should be made to raise riders' awareness of the risk concerns. (line 29-32)
Point 2: Starting close to the end of first paragraph, the sentences are too long and not easily readable in the introduction.
Responses2: Line 44-51 in the original version had been adjusted to make the sentences more concise and easier to understand. As shown in the revisions line 52-57.
Modify2: In China, E-bikes are defined as electric two wheelers with relatively low speeds and weights compared to motorcycles, which can be divided into three types: pedal-type, light motor-type and motor-type, with the maximum design speed of 25km/h, 50km/h and more than 50km/h, respectively. Due to the lack of current management standards, there is no much difference in dynamic performance of pedal-type and light motor-type e-bikes. However, the motor-type e-bikes with a high speed are almost nonexistent in China. (line 52-57)
Point 3: 2nd paragraph is OK but suddenly jumps to SEM at the next paragraph then jump back to data to explain driving behaviors.
Responses3: In the original version, the sequence of the paragraphs is somewhat confused. The third paragraph has been moved to the behind of fifth paragraph. As the existing research on motor vehicle driving behavior should be introduced at first, then the existing research on e-bike riding behavior, and then the structural equation model adopted in this paper. Moreover, the beginning or end of the three paragraphs were adjusted to make the paragraphs better cohesive. As shown in the revisions line 79-81, line 96-99, line 109-110.
Point 4: Lines 125-126 are confusing and does not add a value to paragraph.
Responses4: Lines 125-126 failed to explain the source of the two dimensions of the riding confidence scale well leading to confusion, so it was supplemented and modified. As shown in the revisions line 153-156, line 159-160.
Modify4:① However, according to the definition of confidence in social psychology, the source of confidence is not only the affirmation of technical capacity, but also the affirmation of individual judgment. Especially the judgment of the behavior and surrounding environment based on personal riding experience (line 153-156)
②Therefore, this paper added four questions based on the Wong’s riding confidence scale. The riding confidence scale was dived into two dimensions (line 159-160)
Point 5: Lines 146-148 are confusingly written.
Responses5: Lines 146-148 in the original version showed the preliminary scale of risk perception designed based on existing studies, and combined the exploratory factor analysis results to determine the final scale dimension. However, this arrangement made the structure and layout of the article confusing. Therefore, this sentence was modified to just introduce the original scale. As shown in the revisions line 187-188. Then a new section was added to introduce the results of exploratory factor analysis, as shown in the revisions section 3.2.
Modify5: Therefore, this paper designed risk perception scale, which was divided into two dimensions to evaluate concern degree and danger perception on crashes. (line 187-188)
Point 6: Response to comment: Is this based on the Rundmo's paper?
Responses6: Yes, based on the research of Rundmo, Machin, Yao, et al, this paper improved the structural model of e-bikes risky riding behavior and conducted a more comprehensive study on the latent relationships between internal influencing factors and riding behaviors.
Point 7: Response to comment: when there are mutiple predictors, it is incorrect to use Rsquare (adjusted is more appropriate) to measure the explanatory power.
Responses7: As for structural equation model, basically all reported is R-square,however, considering the limitation of R-square in multiple predictors, the evaluation of structural equations is mainly based on the indices of fitting degree. As all the fitting indices of the model in this study have reached the standard, the model is good.
Point 8: Response to comment: Diagrams are good but you started with somebody else's model or inputs.
Responses8: The preliminary scale design was based on the previous research results. After site observation and analysis from the perspective of psychology, the questionnaire scales were improved. For example, in terms of safety attitude scale, “herd mentality” as a unique characteristic of e-bikers was added. Then the model was built based on the questionnaire analysis results. Therefore, this model was based on previous researchers' models and is different from theirs. Therefore, this research was based on the previous research to form our own model diagrams, which is different from the previous model.
We appreciate for your warm work earnestly, and hope that the correction will meet with approval.
Once again, thank you very much for your comments and suggestions.
Corresponding Author
WANG Tao
Reviewer 2 Report
This is a much improved version; I appreciate the effort made to address the concerns raised and to improve the clarity of the paper.
There are still a number of places where attention is needed to improve the quality of grammar. E.g. sentences shouldn't start with "And"; the plural of "research" is also "research".
Author Response
Dear Reviewer:
Thank you for your comments concerning our manuscript. Those comments are all valuable and very helpful for revising and improving our paper. We have studied comments carefully and have made correction which we hope meet with approval. Revised portion are marked in red in the paper. The main corrections in the paper and the responds to the reviewer’s comments are as flowing:
Point 1: There are still a number of places where attention is needed to improve the quality of grammar. E.g. sentences shouldn't start with "And"; the plural of "research" is also "research".
Responses1: The problems about the listed "And" and " research" had been corrected in the text. And the grammar problems in other places were also modified.
Once again, thank you very much for your comments and suggestions.
Corresponding Author
WANG Tao

Reviewer 3 Report
This manuscript focused on the intrinsic causal relationships between influencing factors and risky riding behavior of e-bike riders. The data regarding individuals’ riding behavior and their intention were obtained using the E-bike Rider Behavior Questionnaire (ERBQ). The SEM was constructed to study the relationships among the four dimensions of "risk perception", "riding confidence", "safety attitude" and "risky riding behavior" of e-bike riders. Some valuable conclusions were drawn, which can provide some useful information for future development of e-bike safety interventions.
However, I have some concerns as follows:
* The survey samples are only from Guilin and Nanning in Guangxi Zhuang Autonomous Region, which are both southern cities and can’t represent the whole China well. The samples from the central, western and northern cities are missing. Possibly a larger sample for multiple cities would make the paper rigorous to be considered by the journal.
* The authors had better list the load coefficients between the observed variables and the initial factors. Such results contribute to a better understanding of the riding behaviors of e-bike riders.
* In Figures 1, 2 and 3, parameters e1, e2, ..., e12, er1, er2, and er3 may represent measurement residuals. The meaning of e1, e2, ..., e12, er1, er2, and er3 is not explained in the text.
* The standardization factor loads of “technical capacity” and “judgement capacity” shown in Figure 3 are in conflict with the description on line 330 of the text. Please verify and correct.
* On line 267 and line 268, “danger level” should be the initial factor, but not the observed variables. Please verify and correct.
* There are some format problems. For instance, the author name is written irregularly on line 84; a reference should be added on line 133; the expression of “five-point Likert scale” is inconsistent in the text; etc. Please check and correct carefully.
Author Response
Dear Reviewer:
Thank you for your comments concerning our manuscript. Those comments are all valuable and very helpful for revising and improving our paper, as well as the important guiding significance to our researches. We have studied comments carefully and have made correction which we hope meet with approval. Revised portion are marked in red in the paper. The main corrections in the paper and the responds to the reviewer’s comments are as flowing:
Point 1: The survey samples are only from Guilin and Nanning in Guangxi Zhuang Autonomous Region, which are both southern cities and can’t represent the whole China well. The samples from the central, western and northern cities are missing. Possibly a larger sample for multiple cities would make the paper rigorous to be considered by the journal.
Responses1: The survey samples are only from Guilin and Nanning, but nowadays the small and medium-sized cities, especially in southern cities, have a higher proportion of e-bike use. Moreover, according to the contrast of Nanning and Guilin samples, found that geographical differences have few impacts on sample data. Therefore, although the samples can’t represent the whole China well, the results will not be very different from other regions of China. In future study, we will consider expanding the sample size to investigate multiple regions
Point 2: The authors had better list the load coefficients between the observed variables and the initial factors. Such results contribute to a better understanding of the riding behaviors of e-bike riders.
Responses2: The load coefficients between the observed variables and the initial factors had been listed. As shown in the revisions Appendix A. Moreover, section 3.2 in the revisions was added to explain the results of Appendix A.
Point 3: In Figures 1, 2 and 3, parameters e1, e2, ..., e12, er1, er2, and er3 may represent measurement residuals. The meaning of e1, e2, ..., e12, er1, er2, and er3 is not explained in the text.
Responses3: In Figures 1, 2 and 3, parameters e1, e2, ..., e12 represented measurement errors, and parameters er1, er2, and er3 represented structure residuals. The meaning of them were explained, as shown in the revisions line 363-367.
Modify3: The (n=1, 2…12) represented the measurement errors, which reflected the unexplainable variances of the measured variables after being estimated. The (n=1, 2, 3) represented the structure residuals, which reflected the unexplainable variances of the structural equation. The standardized value of residuals was below 2.58, indicating that the internal structure of the model had a good fitting degree. (line 363-367)
Point 4: The standardization factor loads of “technical capacity” and “judgement capacity” shown in Figure 3 are in conflict with the description on line 330 of the text. Please verify and correct.
Responses4: clerical error, the standardization factor loads of “technical capacity” and “judgement capacity” were written in reverse. It had been corrected, as shown in the revisions line 414-415.
Modify4: “judgement ability” (0.90) and “technical capacity” (0.68), (line 414-415)
Point 5: On line 267 and line 268, “danger level” should be the initial factor, but not the observed variables. Please verify and correct.
Responses5: clerical error, “danger level” is the initial factor (initial potential variable). It had been corrected. In addition, this sentence had some grammatical problems and had been corrected together. As shown in the revisions line 339-342.
Modify5: First, the standardization factor loading between the initial potential variable "danger level" and the higher-order potential variable "risk perception" was only 0.314 (preferably between 0.4-0.96), so the variable "danger level" should be deleted. (line 339-342)
Point 6: There are some format problems. For instance, the author name is written irregularly on line 84; a reference should be added on line 133; the expression of “five-point Likert scale” is inconsistent in the text; etc. Please check and correct carefully.
Responses6: The format problems listed above had been corrected in the text. As shown in the revisions line 100-101, line 168, and line 161-162, line222-223. Moreover, other format problems had been corrected.
Modify6: ①Wang.[10] also demonstrated that the comprehensive traffic collision rates of e-bikes (line 100-101)
②The risk perception scale was adapted from the study of Rundmo, Machin et al. [16,17].. (line 168)
③Each question was measured on a 5five-point Likert scale. (line 161-162)
④Each question was scored by a Likert five-point Likert scale. (line 222-223)
We appreciate for your warm work earnestly, and hope that the correction will meet with approval.
Once again, thank you very much for your comments and suggestions.
Corresponding Author
WANG Tao
Round 2
Reviewer 1 Report
measurement error or residuals cannot be a single number. your tables are confusing. comments on diagrams are the same as before. When there is interaction, you cannot just multiply their individual coefficients.
Reviewer 3 Report
The doubts I raised for the first revision have been addressed. I accept the authors' responses.